# Outcome After Laparoscopic Compared to Open Interval Debulking Surgery for Advanced Stage Ovarian Cancer: A Systematic Review and Meta-Analysis

**DOI:** 10.3390/cancers17233858

**Published:** 2025-11-30

**Authors:** Jana von Holzen, Franziska Siegenthaler, Noah Locher, Christine Baumgartner, Sara Imboden, Michael David Mueller, Flurina Annacarina Maria Saner

**Affiliations:** 1University of Bern, 3012 Bern, Switzerland; jana.vonholzen@students.unibe.ch; 2Department of Gynaecology and Obstetrics, Bern University Hospital, Faculty of Medicine, University of Bern, 3010 Bern, Switzerlandnoah.locher@insel.ch (N.L.); sara.imboden@insel.ch (S.I.); michel.mueller@insel.ch (M.D.M.); 3Department of Infectious Diseases, Bern University Hospital, Faculty of Medicine, University of Bern, 3010 Bern, Switzerland; christine.baumgartner@insel.ch

**Keywords:** ovarian cancer, minimally invasive surgery, neoadjuvant chemotherapy, interval debulking surgery

## Abstract

This study looked at whether keyhole (minimally invasive) surgery is as safe and effective as open abdominal surgery for women with advanced ovarian cancer who have already received chemotherapy to shrink their tumours. Researchers reviewed 14 studies involving over 16,000 patients. They found that women who had minimally invasive surgery were slightly more likely to have all visible cancer removed and had similar survival rates compared to those who had open surgery. Patients who underwent the less invasive approach also had fewer complications, lost less blood, stayed in the hospital for a shorter time, and could start further chemotherapy sooner. These results suggest that minimally invasive surgery can be a safe and effective option for selected patients. However, more large, well-designed studies are needed to confirm these findings and to identify which women will benefit most from this approach.

## 1. Introduction

Characterized by its aggressive behaviour, ovarian cancer is one of the most lethal gynecological malignancies in women globally, representing a clinical challenge with generally poor prognosis [1]. Due to the non-specific symptoms and lack of reliable screening tools, the tumour is often detected at an advanced stage, which is one of the main reasons for its lethality [2,3]. A radical, complete resection of all visible tumour, is the main goal of ovarian cancer surgery and the most important prognostic factor for patient survival [4,5], along with other clinical determinants and genomic and immune features of the tumour [6,7,8].

Over the last two decades, neoadjuvant chemotherapy followed by interval debulking surgery has gained importance as a viable alternative to primary debulking surgery for advanced stage FIGO IIIC and IV ovarian cancer. Neoadjuvant treatment has been shown to reduce perioperative morbidity by achieving similar oncological outcomes compared to primary debulking, without compromising survival [9,10,11]. This strategy is particularly beneficial for women with heightened perioperative risk, relevant comorbidities, or in situations where downstaging of the tumour is necessary to achieve complete cytoreduction.

Current standard for both primary and interval debulking surgery is an open surgery through midline laparotomy. With the increasing adoption of minimally invasive techniques, recent studies found potential benefits of a laparoscopic approach in primary cytoreductive surgery for ovarian cancer [12,13,14], including reduced invasiveness, decreased blood loss, shorter hospital stays, and fewer postoperative adhesions. Similar advantages were seen in patients undergoing laparoscopic interval debulking surgery after neoadjuvant chemotherapy [15,16,17,18]. However, the minimally invasive approach remains controversial due to a potential underestimation of the intra-abdominal tumour load with higher risk for incomplete cytoreduction and potential complications such as trocar metastases and ovarian tumour rupture in early-stage disease [19].

Therefore, to date, midline laparotomy has remained the standard of care for ovarian cancer surgery, whereas a minimally invasive approach is mainly used in early FIGO stages, fertility-sparing surgeries, and to evaluate optimal cancer resectability before primary or interval debulking surgery [20,21]. It remains unclear, however, if a laparoscopic debulking surgery is an oncologically safe alternative in patients with reduced tumour burden after neoadjuvant chemotherapy.

This systematic literature review and meta-analysis compares the efficacy, safety, and oncological outcomes of laparoscopic versus open interval debulking surgery in patients with advanced-stage ovarian cancer (FIGO-stage III-IV). Compared to previous reviews, this meta-analysis is the first to incorporate the results of the only randomized controlled trial on this topic to date and addresses additional outcomes in a substantially larger patient population.

## 2. Materials and Methods

### 2.1. Search Strategy

This systematic review and meta-analysis followed a detailed study protocol that complied with the Preferred Reporting Items for Systematic Reviews and Meta-Analyses (PRISMA) guidelines and has been registered in the PROSPERO database (CRD42024524725). No funding sources or conflicts of interest were declared.

A systematic search of the literature was conducted using the key words “ovarian cancer”, “laparoscopy”, “laparotomy”, and “neoadjuvant chemotherapy” in the Ovid/Medline, Pubmed, and Cochrane databases. To locate supplementary studies of potential relevance, references cited within the included papers were screened for eligibility. The literature search was limited to articles published up to 15 February 2025.

### 2.2. Study Selection Criteria

The study selection was performed independently by two investigators (JvH, FAMS), who screened titles, abstracts, and full-text articles for eligibility based on predefined inclusion and exclusion criteria. Inclusion criteria were defined as follows: (1) randomized trial or observational study, (2) population of patients with advanced-stage (FIGO stage III and IV) ovarian, fallopian tube, or primary peritoneal cancer undergoing neoadjuvant chemotherapy, (3) subgroup undergoing minimally invasive (laparoscopic or robot-assisted) interval debulking surgery, and (4) subgroup undergoing laparotomy for interval debulking.

Exclusion criteria were as follows: (1) duplicate publications, abstracts, letters, editorials, review, and expert opinion, (2) recurrent ovarian cancer, pregnant women with ovarian cancer; (3) second-look laparotomy/secondary surgical cytoreduction, laparotomy and laparoscopy for diagnostic reasons only; and (4) comparison of primary versus interval debulking surgery.

Using these search criteria, comparative studies that included patients with advanced-stage ovarian cancer FIGO stage III-IV that set side by side groups of patients undergoing interval debulking surgery with either laparoscopy or laparotomy were considered.

### 2.3. Definition of Outcomes

Surgical resection status was the primary outcome. Complete surgical cytoreduction with no macroscopic tumour left was classified as R0 resection; R1 and R2 resection were defined as visual residual disease with a diameter of 0.1–1 cm (R1) or >1 cm (R2), respectively. Optimal cytoreduction was defined as either R0 or R1 resection.

Secondary outcomes included overall survival, progression-free survival, perioperative outcomes including blood loss and operative time, and postoperative outcomes including complications, length of hospital stay, and delay of adjuvant chemotherapy.

### 2.4. Data Extraction and Statistical Analysis

The following data of included studies were collected: year of publication, study design and duration, FIGO stage, number of patients per intervention group, mean age, mean number of neoadjuvant chemotherapy cycles, details of surgery (rate of laparoconversion, operation time, blood loss, intraoperative complications, in the minimally invasive cohort: robotic-assisted versus laparoscopic surgery), surgical resection rate, postoperative course (postoperative complications, duration of hospital stay, days to adjuvant therapy, readmission rate), mean follow up in months, recurrence rate, progression-free survival, and overall survival. To obtain missing data, corresponding authors of the studies concerned have been contacted, with one feedback.

We conducted a meta-analysis using a random-effects model to analyze both binary and continuous outcomes, as the studies combined exhibit variations in terms of methodology and population characteristics.

Mean values for continuous outcomes were calculated using a weighted average based on the number of patients in each cohort. The relative risk with a 95% confidence interval was calculated for binary outcomes, such as resection rate (R0- and R0R1-resection), intra- and post-operative complications, and readmission and recurrence rates, using the raw data extracted from the studies. Outcome estimates of risk ratios (RR), hazard ratios (HR), and standardized mean differences (SMD), with the corresponding 95% confidence intervals (CI), respectively, were combined, and a pooled estimate of the effect size was generated weighting the individual study effect estimates by their inverse variance.

Due to limited availability of reported values for time-to-event and continuous outcomes including blood loss, operative time, and length of stay, indirect techniques had to be used to estimate the different effect sizes. Missing values such as the logarithm of the extracted hazard ratios and the logarithm of its variance were computed using the Action “Calculator” of the RevMan Software (Version 5.4) to obtain the overall pooled hazard ratio for time-to-event outcomes from raw data like standard deviation and mean values. No reconstruction from Kaplan–Meier curves or Tierney-type methods was performed. Moreover, under the assumption of a normal distribution, the missing standard deviations of continuous outcomes were estimated using the given interquartile ranges or published 95% confidence intervals. The following formulas were applied to do so:Standard deviation = √n × (Upper CI − Lower CI)/3.92Standard deviation = IQR/1.35Variance = ((Upper CI − Lower CI)/3.92)^2^

Data extraction and statistical analysis were performed using the Review Manager Software (The Cochrane Collaboration, 2020, Version 5.4).

### 2.5. Subgroup and Sensitivity Analysis

Given the considerable heterogeneity observed among the studies, exploratory subgroup and sensitivity analyses were conducted. Subsequent meta-analyses were performed after restricting the analysis to studies with a lower risk of bias and after excluding studies involving robotic MIS. Additionally, a comparison of fixed- and random-effects models was undertaken in RevMan software (version 5.4).

### 2.6. Heterogeneity and Risk of Bias Assessment

Homogeneity and heterogeneity among studies were assessed using the Higgins I^2^-test, the χ2-test, and the restricted maximum likelihood method, all calculated using RevMan software (version 5.4). The risk of bias was assessed and evaluated according to study design. Randomized controlled trials were assessed with Cochrane RoB 2, whereas all non-randomized studies were assessed using ROBINS-I.

### 2.7. Assessment of Evidence Certainty

The GRADEpro Guideline Development Tool platform (https://gradepro.org accessed on 27 November 2025) was used to evaluate the strength of the evidence through a non-contextualized approach, taking into account the risk of bias, inconsistency of the effect, indirectness, imprecision, and publication bias. Outcomes based exclusively on observational studies were initially assigned low certainty and further downgraded in cases of significant concerns. For outcomes where the randomized trial contributed alongside observational studies, both the RoB 2 and ROBINS-I assessments were considered when judging the risk-of-bias domain.

In accordance with the journal’s guidelines, we will provide our data for independent analysis by the selected Editorial Team for the purposes of additional data analysis or for the reproducibility of this study in other centres if such is requested.

## 3. Results

Overall, 535 studies were identified. After exclusion of duplicates, 368 abstracts and titles were screened. Of those, 80 full text articles were assessed for eligibility. Fourteen trials, published between 2015 and 2024, met the eligibility criteria and were included in this study (Figure 1).

### 3.1. Study Characteristics

Overall, 16,578 patients were included: 4310 in the minimally invasive group, including 1164 robotic-assisted surgeries, and 12,268 women with laparotomy after neoadjuvant chemotherapy (Table 1). The mean rate of laparoconversion across all studies was 8.52%. Eleven studies were of retrospective nature; one randomized-controlled trial was identified [22].

Mean age of patients was similar in the minimally invasive (64.8 years) and open groups (63.7 years, Appendix A). When comparing the treatment intensity of the laparotomy and minimally invasive cohort, the former received an average of 3.8 cycles of neoadjuvant chemotherapy compared to 4.3 cycles before minimally invasive surgery, as reported in seven studies. Median follow-up was slightly shorter in the laparoscopic cohort compared to the open cohort (28.9 vs. 30.2 months).

### 3.2. Resection Rate

Overall, results of surgical resection status were indicated in 12 studies. The rate of complete cytoreduction was significantly higher after minimally invasive surgery (RR = 1.12; 95% CI = [1.01, 1.23]; *p* = 0.03, Figure 2A) and optimal cytoreduction to ≤1 cm residual disease (RR = 1.07; 95% CI = [1.01, 1.14]; *p* = 0.03, Appendix A) as reported in eight studies. Substantial heterogeneity was found in both analyses (complete cytoreduction I^2^ = 68%, *p* = 0.004; optimal cytoreduction: I^2^ = 71%, *p* = 0. 002).

### 3.3. Survival Data

The meta-analysis of seven studies showed no significant differences in overall survival (HR = 0.81; 95% CI [0.64, 1.04]; *p* = 0.10) between the two groups (Figure 2B, Table 2). Again, substantial heterogeneity was observed among the included studies (I^2^ = 92%, *p* < 0.001). The HR for progression-free survival (data from four studies) was 0.67 (95% CI [0.48, 0.94]; *p* = 0.02; I^2^ = 55%; *p* = 0.07), indicating a significant benefit for the minimally invasive group in comparison to the open group (Figure 2C).

Due to substantial heterogeneity and risk of bias in some studies, subgroup analyses were performed. Restricting the analysis to studies with low to moderate risk of bias, or to studies with robotic surgeries only in the MIS cohort, the pooled estimates for R0 resection and survival endpoints changed marginally, and MIS remained associated with higher complete debulking rates consistently (Appendix A).

### 3.4. Secondary Outcomes

The relative risk for postoperative complications was significantly lower after laparoscopy (RR = 0.50; 95% CI [0.33, 0.76]; *p* < 0.001, Figure 3A), with moderate heterogeneity between the studies (I^2^ = 49%, *p* = 0.07). Across all 14 studies included, both mild (grade 1–2) and severe (grade 3–4) postoperative complications occurred more often after open compared to laparoscopic surgery (grade 1–2: 26.7% vs. 15.5%; grade 3–4: 14.2% vs. 7.2%) according to the Clavien-Dindo classification [36]. Intraoperative complication rates were indicated in five studies and did not differ significantly between both groups (RR = 0.96; 95% CI [0.47, 1.98]; *p* = 0.91, heterogeneity: I^2^ = 0%, *p* = 0.78; Appendix A).

Mean blood loss during minimally invasive surgery was significantly lower compared to laparotomy in five studies (SMD = −0.58; 95% CI [−0.82, −0.35]; *p* < 0.001; without important heterogeneity), corresponding to an approximate absolute reduction of 160 mL (MIS 165 mL vs. LPT 325 mL). Time of surgery was significantly longer with a minimally invasive approach (SMD = 0.50; 95% CI [0.11, 0.89]; *p* = 0.001, Figure 3B) with a considerable heterogeneity between the studies (I^2^ = 51%, *p* = 0.09). Hospital stay was significantly shorter after laparoscopy (SMD = −0.79; 95% CI [−1.06, −0.52]; *p* < 0.001, Figure 3C), equivalent to an approximate reduction of about 2 days (3 vs. 5 days).

Overall, a faster transition from surgical intervention to adjuvant chemotherapy was detected in the minimally invasive compared to the open group (mean 25.3 vs. 33.1 days; Appendix A). In terms of readmission within 30 days there was no significant difference between the two groups (RR = 0.82; 95% [0.41, 1.64]; *p* = 0.42; heterogeneity: I^2^ = 57%, *p* = 0.07). There were significantly fewer recurrences detected after minimally invasive compared to open surgery in the meta-analysis of three studies (RR = 0.86; 95% CI [0.83, 0.89]; *p* = 0.02; heterogeneity: I^2^ = 0%, *p* = 0.98, Appendix A).

**Table 2 cancers-17-03858-t002:** Complete cytoreduction and survival after minimally invasive compared to open interval debulking surgery.

Author	Year	R0-Resection		*p*-Value	Overall Survival, Months		*p*-Value	HR	Progression-Free Survival, Months		*p*-Value	HR
		LPT, n (%)	MIS, n (%)		LPT	MIS			LPT	MIS		
Abitbol [23]	2019	9 (40.91)	47 (82.46)	NR	Median: 37.9 ± 9.8	Median: 47.2 ± 9.8	0.02	0.5 ± 0.1	Median: 11.9 ± 1.2	Median: 20.6 ± 2.4	0.002	0.5 ± 0.1
Brown [24]	2019	44 (42.31)	32 (60.38)	0.02	Median: 35	Median: 37	0.74	0.92 (95%CI 0.54–1.55)	Median: 29	Median: 27	0.45	0.83 (95%CI 0.51–1.34)
Brown J [25]	2021	135 (46)	80 (66)	<0.001	Median: 36.7	Median: 40.9	0.5	NR	Median: 15.1	Median: 18.2	0.051	NR
Davidson [26]	2018	145 (59.18)	20 (76.92)	NR	NR	NR	NR	NR	NR	NR	NR	NR
Favero [27]	2015	11 (100)	10 (100)	NR	NR	NR	NR	NR	Mean: 20.5 ± 3.7	Mean: 13.3 ± 2.2	0.29	NR
Gueli Alletti [28]	2016	62 (95.4)	29 (96.67)	NR	NR	NR	NR	NR	Median: 12	Median: 18	0.027	NR
Jorgensen [29]	2023	780 (59.17)	873 (64.38)	0.01	Median: 41 (95%CI 38.7–43.5)	Median: 46.7 (95%CI 44.7–48.6)	<0.01	0.86 (95%CI 0.79–0.94)	NR	NR	NR	NR
Lecointre [30]	2022	NR	NR	NR	26.3 (95%CI 21.7–31.7)	Median: 23.1 (95%CI 15.7–29.7)	0.17	0.45 (95%CI 0.19–0.95)	Median: 12	Median: 14.8	0.057	0.71 (95%CI 0.27–1.88)
Melamed [31]	2017	789 (49.47)	130 (46.93)	0.29	Median: 26.3 (95%CI 21.7–31.7)	Median: 33.8 (95%CI 31.9–40.6)	NR	1.09 (95%CI 0.93–1.28)	NR	NR	NR	NR
Pereira [32]	2022	20 (86.96)	7 (100)	0.31	Median: 37.6 (95%CI 36.1–39.6)	NR	NR	NR	NR	NR	NR	NR
Persenaire [33]	2022	1567 (35.5)	329 (37.3)	0.59	Median: 35.1	Median: 36.6	0.22	0.94 (95%CI 0.86–1.04)	NR	NR	NR	NR
Pomel [34]	2021	NR	31 (96.9)	NR	NR	NR	NR	NR	NR	NR	NR	NR
Rauh-Hain [22]	2024	39 (83)	42 (88)	NR	NR	NR	NR	NR	NR	NR	NR	NR
Zhang [35]	2021	26 (52)	27 (62.79)	0.4	Mean: 38.2	Mean: 35.6	0.7	1.02 (95%CI 0.54–1.93)	Mean: 15.4	Mean: 16.7	0.7	0.86 (95%CI 0.53–1.40)

Abbreviations: R0, resection to no visible residual disease; MIS, minimally invasive surgery; LPT, laparotomy; HR, hazard ratio; CI, confidence interval; n, number of patients; (%), percentage of patients; NR, not reported.

### 3.5. Risk of Bias and Sensitivity Analysis

Overall, the analysis included studies with a moderate to high risk of bias for all outcomes, which was mainly due to the observational nature of the studies and the amount of missing data (Appendix A). The certainty of evidence as assessed by GRADE was moderate to high for the primary outcome, and mostly very low to low for the secondary outcomes (Appendix A). Sensitivity analysis comparing fixed- and random-effects models showed consistent results for primary outcomes (R0 resection rate with fixed-effects model RR = 1.08; 95% CI [1.04, 1.12]; *p* < 0.001; random-effects model RR = 1.12; 95% CI [1.01, 1.23]; *p* = 0.03; Appendix A).

## 4. Discussion

### 4.1. Summary of Main Results

This systematic review and meta-analysis of overall 14 studies found a higher rate of complete cytoreduction and progression-free survival, as well as a lower risk of postoperative complications, lower intraoperative blood loss, and shorter hospital stay in women undergoing minimally invasive surgery after neoadjuvant treatment for advanced-stage ovarian cancer, compared to open surgery. There was no difference in overall survival, the risk for intraoperative complications, and the rate of readmission between both cohorts; the duration of the surgery was longer in patients with laparoscopy. However, risk of bias was moderate to high in most studies with a potential selection bias for MIS in patients with a good response to chemotherapy in cohort studies. Certainty of evidence for most outcomes was low, highlighting the necessity for larger prospective studies and future randomized interventional studies to validate the use of minimally invasive surgery and identify patients with advanced stage ovarian cancer who may benefit from a minimally invasive approach after neoadjuvant chemotherapy.

### 4.2. Results in the Context of the Published Literature

Our results both confirm and challenge the findings of prior systematic reviews investigating the outcomes of minimally invasive compared to open interval debulking in advanced-stage ovarian cancer, albeit in a substantially larger population of overall 16,578 patients [37,38,39]. Newly published studies were included [22,29,30,33], and additional outcomes such as mean blood loss and delay to adjuvant chemotherapy were investigated.

### 4.3. Surgical Resection Status

This meta-analysis found significantly higher rates of complete cytoreduction to no macroscopic disease (R0 resection) during laparoscopy compared to laparotomy. Also, the rate for an either R0 or R1 resection—previously considered as optimal cytoreduction—was more frequent in patients undergoing minimally invasive surgery. Of the included studies, only two reported a significant difference in R0 resection favouring laparoscopy [24,29], with no significant difference in the other trials. An earlier, smaller meta-analysis comprising four studies found no significant difference in the rate complete (R0) or optimal (R0/R1) cytoreduction between laparoscopic and open surgery [38], which can be explained by an increase in surgical expertise and improved selection criteria for minimally invasive surgery over time. In the course of the ongoing randomized controlled LANCE study, Rauh-Hain et al. conducted a trial feasibility study, in which they found a difference in gross resection rates between the minimally invasive (88%) and laparotomy (83%) groups of 4.5 percentage points [22].

### 4.4. Patient Survival

Our analysis shows similar overall survival in both groups, with a trend towards a survival benefit for patients with minimally invasive surgery. Jorgensen et al. reported a significantly longer overall survival of 46.7 months compared to 41 months after laparoscopy (*p* < 0.01) in their study including 4042 patients, further supported by Abitbol et al. who found a significantly longer overall survival (+9.3 months) after laparoscopy in a small cohort of 79 patients [23,29]. Consistent with previous works [37,38], progression-free survival was significantly longer after minimally invasive surgery in our analysis, indicating a potential benefit of this surgical approach after neoadjuvant treatment in selected patients. A significantly longer progression-free survival after laparoscopy was found in two of the studies included in this meta-analysis [23,28]. However, survival data from comparative cohort studies need to be interpreted with caution regarding the high risk of confounding by indication of non-randomized, retrospective trials, with women showing a favourable response to chemotherapy and a low tumour burden left being more likely to undergo minimally invasive surgery. Although prospective randomized investigations are required to determine overall and progression-free survival, our data indicate that both the minimally invasive and the open approach may result in comparable survival rates.

### 4.5. Intraoperative Course

Consistent with previous studies [37], minimally invasive procedures lasted significantly longer compared to open interval debulking surgeries in this study, which is not surprising given the higher complexity of a minimally invasive approach. It is worth noting that the beginning and end of the calculation of surgery time were not clearly defined and inconsistent between the different studies included. Intraoperative complication rates showed no difference between both groups. However, the laparoscopic cohort had a significantly lower mean blood loss, which is consistent with findings also in surgery for other cancer types including cervical or colorectal cancer [40,41].

### 4.6. Postoperative Complications and Length of Stay

The risk of postoperative complications in the minimally invasive group was half compared to the group undergoing laparotomy. This substantial risk reduction was mainly driven by two of the included studies [25,35], in which patients undergoing minimally invasive surgery had a lower surgical complexity and fewer wound complications, ileus, and blood transfusions. The recently published randomized controlled trial of Rauh-Hain et al. corroborates these findings, with a postoperative complication rate of 8% in the minimally invasive and 26% in the open cohort [22], suggesting an enhanced postoperative recovery after laparoscopy. Our data contradict the findings of two previous meta-analyses with fewer patients, where no impact of the mode of surgery on postoperative complications was found [38]. The benefit of minimally invasive surgery on postoperative complications is further reflected in the significantly shorter hospital stay after laparoscopic surgeries found in this study.

Several of the included studies contribute to these findings with all having significantly fewer days of hospital stay after laparoscopy [22,29,30,33] and a lower estimated blood loss [37]. Thus, it can be assumed that the minimally invasive procedure promotes faster recovery and facilitates a swifter return to the patient’s daily life. This hypothesis is supported by the findings of two previous meta-analyses, which showed a significantly reduced duration of hospitalization compared to patients with laparotomy [37,38].

Data on the delay between surgery and initiation of adjuvant chemotherapy was scarce among the included studies. We found a shorter mean interval to chemotherapy after minimally invasive versus open surgery, probably due to a generally quicker recovery [13]. This potential advantage of the laparoscopic approach in the early initiation of postoperative chemotherapy may contribute to improved treatment outcomes.

### 4.7. Current Evidence

To the best of our knowledge, this is the most recent meta-analysis comparing the minimally invasive to the open debulking surgery after neoadjuvant chemotherapy for advanced stage ovarian cancer. A prospective randomized phase III trial, the LANCE (Laparoscopic Cytoreduction After Neoadjuvant Chemotherapy) trial, is now ongoing, and the data from their pilot phase has already been incorporated in our meta-analysis [22]. The results of this trial will offer valuable insights and will be relevant to validate and potentially challenge the findings of this work.

This systematic review and meta-analysis reveal a general lack of standardization in the decision-making process for choosing between open or laparoscopic surgery. Most of the studies reported that the treating physician decided based on clinical, radiological, and serological markers [17,24,35]. Only two of the observational studies used propensity-matched cohorts to mitigate confounding by indication [29,30], and only one randomized controlled trial was included in our meta-analysis [22], suggesting that the minimally invasive and open groups may not be directly comparable.

Inconsistencies were also observed in diagnostic laparoscopy to evaluate the chance of primary debulking, which was routinely performed in one study [30] but not in the cohort of Brown et al. [24]. These approaches are individualized depending on the institute and clinic, which presents a challenge for interpretation of pooled data from meta-analyses and systematic reviews.

### 4.8. Strengths and Weaknesses

Limitations of this systematic review and meta-analysis consist of the paucity of randomized controlled trials on this topic. The existing, mainly retrospective studies show substantial variations in study design, patient populations, and methodologies, such as chemotherapy regimen, surgical technique, and postoperative care, which may lead to discrepancies in specific outcomes reported in the studies. The substantial heterogeneity found between studies, and the potential selection of patients with a favourable response to chemotherapy to be treated with minimally invasive surgery may have confounded our results. In addition, missing data, often selectively reported, complicated the analysis. Considerable efforts were made to obtain these missing data and the authors of the studies concerned were contacted, but to no avail. In addition, there is an ongoing shift toward R0/CC0 as the preferred definition of optimal debulking, whereas most included studies in this meta-analysis reported cytoreduction either as “R0 vs. R0/R1” or combined R0 and R1 into a single “optimal” category. As a result, we were not able to stratify consistently by R0/CC0 versus R1 or CC1, which should be reported separately in future trials.

However, the reproducibility of previous positive results in a substantially larger population of patients and the analysis of additional, clinically relevant outcomes such as blood loss during surgery and delay to adjuvant treatment are clear strengths of this meta-analysis. The consistent results across multiple subgroup analyses further strengthen the robustness of the primary findings, representing an important strength of this study.

### 4.9. Implications for Practice and Future Research

This meta-analysis highlights the potential benefits of laparoscopic interval debulking surgery compared to laparotomy. Overall, these results suggest that minimally invasive interval debulking surgery in advanced ovarian cancer is an effective and safe option for selected patients with advanced-stage ovarian cancer after neoadjuvant chemotherapy. It is crucial to define clinical markers that help identify patients who would derive maximum benefit from a laparoscopic approach and to standardize patient selection.

Given the higher chance of complete cytoreduction, fewer perioperative complications, and longer progression-free survival seen in the patients after minimally invasive surgery, this approach may be a valuable option in patients showing a favourable response to neoadjuvant chemotherapy. These findings should be validated in future multicentric randomized-controlled trials with awaited results of the LANCE trial further shedding light on this important topic.

## 5. Conclusions

This study provides evidence suggesting that laparoscopic interval cytoreduction is an oncologically safe alternative with a higher rate of complete surgical cytoreduction, similar overall survival and reduced perioperative complication rate compared to open surgery in carefully selected patients with advanced-stage ovarian cancer. However, prospective randomized homogeneous studies are needed to confirm and substantiate these findings. Moreover, the results emphasize the need for future research into clinical markers that can effectively identify patients who will profit the most from a minimally invasive approach.

## Figures and Tables

**Figure 1 cancers-17-03858-f001:**
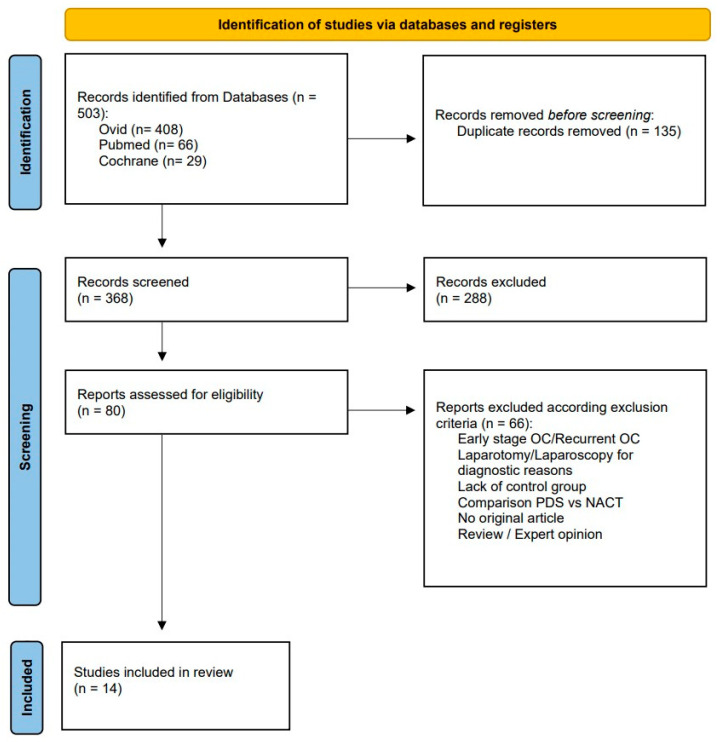
Study selection process (PRISMA Flowchart).

**Figure 2 cancers-17-03858-f002:**
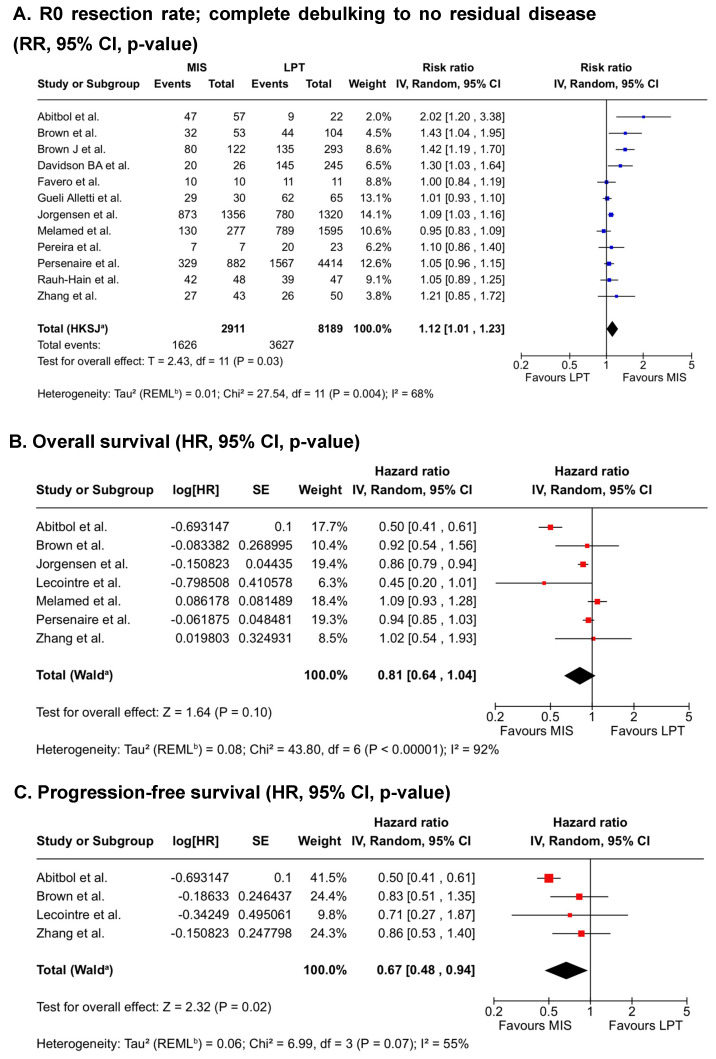
Meta-analysis and forest plots of primary outcomes after minimally invasive compared to open interval debulking surgery [22,23,24,25,26,27,28,29,30,31,32,33,35]. (**A**). Rate of complete cytoreduction after neoadjuvant chemotherapy. (**B**). Overall survival. (**C**). Progression-free survival.Abbreviations: R0, resection to no visible residual disease; RR, risk ratio; HR, hazard ratio; CI, confidence interval; MIS, minimally invasive surgery; LPT, laparotomy; IV, inverse variance; SE, standard error; REML, restricted maximum likelihood, HKSJ Hartung-Knapp-Sidik-Jonkman method; Wald, Wald-type method; I^2^, proportion of variation due to heterogeneity; Tau^2^, between-study variance; df, degrees of freedom; Chi^2^ = Cochran’s Q statistic; weight, contribution of each study to the pooled effect; Test for overall effect Z, Z-value used to test significance. Footnotes: (**A**) ^a^ CI calculated by Hartung-Knapp-Sidik-Jonkman method, ^b^ Tau^2^ calculated by Restricted Maximum-Likelihood method; (**B**) ^a^ CI calculated by Wald-type method, ^b^ Tau^2^ calculated by Restricted Maximum-Likelihood method; (**C**) ^a^ CI calculated by Wald-type method, ^b^ Tau^2^ calculated by Restricted Maximum-Likelihood method. Symbol legend: ■ individual study effect size (weight indicated by square size); — 95% confidence interval of each study; ◆ pooled effect estimate with its 95% CI; │ line of no effect.

**Figure 3 cancers-17-03858-f003:**
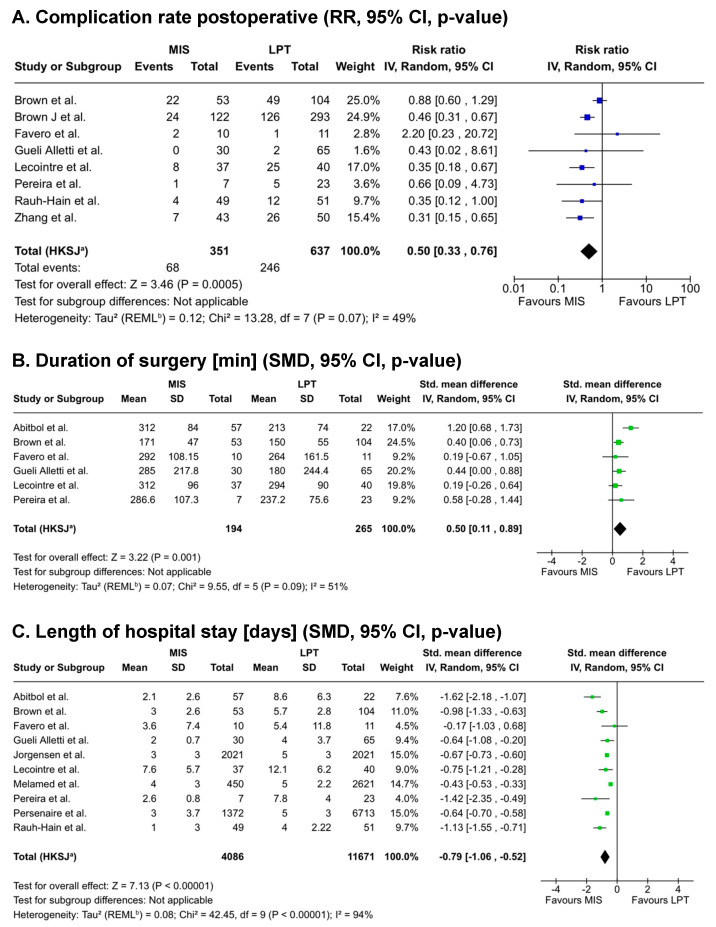
Meta-analysis of secondary perioperative outcomes after minimally invasive compared to open interval debulking surgery [22,23,24,25,27,28,29,30,31,32,33,35]. (**A**). Risk ratio (RR) of postoperative complications. 95% confidence interval and *p*-values are indicated. (**B**). Mean duration of surgery in minutes. (**C**). Length of hospital stay in days after interval debulking surgery depending on type of surgery. Abbreviations: RR, risk ratio; SD, standard deviation; SMD, standardized mean difference; MIS, minimally invasive surgery; LPT, laparotomy; IV, inverse variance; SD, standard deviation; REML, restricted maximum likelihood, HKSJ Hartung-Knapp-Sidik-Jonkman method; I^2^, proportion of variation due to heterogeneity; Tau^2^, between-study variance; df, degrees of freedom; Chi^2^ = Cochran’s Q statistic; Weight, contribution of each study to the pooled effect; Test for overall effect Z, Z-value used to test significance. Footnotes: ^a^ CI calculated by Hartung-Knapp-Sidik-Jonkman method; ^b^ Tau^2^ calculated by Restricted Maximum-Likelihood method. Symbol legend: ■ individual study effect size (weight indicated by square size); — 95% confidence interval of each study; ◆ pooled effect estimate with its 95% CI; │ line of no effect.

**Table 1 cancers-17-03858-t001:** Study characteristics.

Author	Year	Study Design	Duration of Study	FIGO Stage	Patients	Type of Intervention			Laparoconversion, n (%)
					Total	LPT	MIS	Robotic	
					n	n (%)	n (%)	n (%)	
Abitbol [23]	2019	Retrospective	2008–2014	III–IV	79	22 (27.85)	57 (72.15)	57 (100)	6 (10.53)
Brown [24]	2019	Retrospective	2006–2017	III–IV	157	104 (66.24)	53 (33.76)	-	9 (17)
Brown J [25]	2021	Retrospective	2008–2018	III-IV	415	293 (70.6)	122 (29.4)	78 (63.93)	38 (31.15)
Davidson [26]	2018	Retro-prospective	2000–2017	III-IV	272	245 (90.07)	27 (9.93)	-	24 (47.06)
Favero [27]	2015	Prospective	2011–2014	IIIC–IVA	21	11 (52.38)	10 (47.62)	-	0 (0)
Gueli Alletti [28]	2016	Retrospective	2013–2014	III–IV	95	65 (68.42)	30 (31.58)	-	NR
Jorgensen [29]	2023	Retrospective	2013–2018	IIIC-IV	4042	2021 (50)	2021 (50)	864 (42.75)	208 (10.29)
Lecointre [30]	2022	Retrospective	2009–2019	III-IV	77	40 (51.95)	37 (48.05)	-	1 (2.7)
Melamed [31]	2017	Retrospective	2010–2012	IIIC–IV	3071	2621 (85.35)	450 (14.65)	122 (27.11)	72 (16)
Pereira [32]	2022	Retrospective	2012–2013	IIIC-IV	30	23 (76.67)	7 (23.33)	-	0 (0)
Persenaire [33]	2022	Retrospective	2010–2016	IIIC-IV	8085	6713 (83)	1372 (17)	-	183 (2.3)
Pomel [34]	2021	Prospective	2013–2016	IIIC-IV	41	9 (21.95)	32 (78.05)	-	3 (9.38)
Rauh-Hain [22]	2024	RCT	2020–2023	IIIC-IV	100	51 (51)	49 (49)	-	6 (12.5)
Zhang [35]	2021	Retrospective	2011–2018	III–IV	93	50 (53.76)	43 (46.24)	43 (100)	5 (11.63)

Abbreviations: LPT, laparotomy; MIS, minimally invasive surgery; RCT, randomized controlled trial; n, number of patients; (%), percentage of patients.

## Data Availability

The data presented in this study are available on request from the corresponding author.

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
