# Peer review of "Outcome After Laparoscopic Compared to Open Interval Debulking Surgery for Advanced Stage Ovarian Cancer: A Systematic Review and Meta-Analysis"

_cancers, 2025, doi:10.3390/cancers17233858_

Round 1
Reviewer 1 Report
Comments and Suggestions for Authors
This manuscript addresses a clinically important and timely topic: the comparative outcomes of minimally invasive versus open interval debulking surgery after neoadjuvant chemotherapy in advanced ovarian, tubal, and primary peritoneal cancer. The systematic review is generally well designed, the search strategy is clearly outlined, and the inclusion of both large retrospective cohorts and the recent randomized trial substantially increases the sample size and contemporary relevance. The main findings – higher R0/R1 cytoreduction rates, improved progression-free survival, and favorable perioperative outcomes in the minimally invasive surgery (MIS) group, without a clear overall survival disadvantage – will be of interest to gynecologic oncologists and may inform future prospective trials.
At the same time, several methodological aspects require further clarification and a more cautious interpretation of the pooled results. First, the potential for confounding by indication and selection bias needs to be more centrally discussed and, where possible, quantified. In most included studies, surgical approach was not randomized but chosen based on clinical, radiological, and serological parameters, which likely means that fitter patients with lower tumour burden and better chemotherapy response were preferentially selected for MIS. Only a minority of studies applied propensity-score matching or other robust adjustment methods. It would be helpful to make this explicit in the Results and Discussion, and to more clearly acknowledge that the apparent advantages in R0 rates and PFS may at least partly reflect patient selection rather than a purely surgical approach effect.
Second, heterogeneity is substantial for several key outcomes (e.g., cytoreduction status, survival endpoints), yet its sources are only briefly mentioned. Where feasible, additional or more explicit subgroup/sensitivity analyses would strengthen the manuscript: for example, restricting the analysis to studies with propensity-score matching or low risk of bias, separating robotic from conventional laparoscopy, stratifying by study design (RCT vs. retrospective), or exploring the impact of surgical complexity and institutional volume. Even if such analyses are limited by the available data, a more detailed narrative discussion of heterogeneity and its implications for the certainty of pooled estimates would be valuable.
Third, the description and application of risk-of-bias and certainty-of-evidence tools could be clarified and better integrated into the interpretation of findings. It should be clearly stated which tool was used for randomized versus observational studies and how this influenced the GRADE rating for each outcome. Since many outcomes are ultimately rated as low or very low certainty, the language in the Abstract and Conclusions should be aligned with these levels; for example, phrasing the main message as “the available evidence suggests that MIS may be an oncologically safe alternative in selected patients, but the certainty of evidence is low and residual confounding cannot be excluded” would better reflect the current data.
Fourth, the statistical methods would benefit from a more explicit description of how hazard ratios and standard deviations were derived when not directly reported. Indicating whether time-to-event data were estimated from Kaplan–Meier curves (e.g., using Tierney-type methods), specifying the formulas used to approximate SD from IQR or ranges, and briefly summarizing any sensitivity analyses (e.g., removing very large database studies, comparing fixed- versus random-effects models) would enhance reproducibility and allow readers to better judge the robustness of the results.
Finally, there are a few minor points that can improve clarity and readability. Some typographical errors and small inconsistencies in terminology should be corrected (e.g., “minimally invasive” → “minimally invasive”, “Wald-tpye” → “Wald-type”), and key definitions such as R0/R1 cytoreduction could be reiterated in table footnotes for quick reference. For the perioperative outcomes presented as standardized mean differences, it would help clinicians if the text and/or figure legends also highlight the approximate absolute differences in blood loss and length of stay. If available, a brief comment on histologic subtypes and their distribution (e.g., predominance of high-grade serous histology) would also help readers understand the generalizability of the findings.
Overall, this is a well-conceived and clinically relevant meta-analysis. With a more explicit treatment of selection bias and heterogeneity, a closer alignment between the statistical findings and the GRADE-based certainty of evidence, and minor refinements to the wording and presentation, the manuscript will provide a more balanced and informative contribution to the field.
Comments on the Quality of English LanguageThe overall quality of English is good, and the manuscript is generally clear and readable. The scientific message can be understood without difficulty, and the structure of the Introduction, Methods, Results, and Discussion is appropriate. Nevertheless, a number of sentences are rather dense and could be streamlined for easier reading, and there are occasional minor typographical and grammatical errors (e.g., misspellings such as “minimally invasie”, small inconsistencies in hyphenation, and a few punctuation issues). I would therefore recommend a careful proofreading and, if possible, a light professional language edit to further polish the text and improve fluency, but the current level of English does not obscure the scientific content.
Author Response
Comm01: At the same time, several methodological aspects require further clarification and a more cautious interpretation of the pooled results. First, the potential for confounding by indication and selection bias needs to be more centrally discussed and, where possible, quantified. In most included studies, surgical approach was not randomized but chosen based on clinical, radiological, and serological parameters, which likely means that fitter patients with lower tumour burden and better chemotherapy response were preferentially selected for MIS. Only a minority of studies applied propensity-score matching or other robust adjustment methods. It would be helpful to make this explicit in the Results and Discussion, and to more clearly acknowledge that the apparent advantages in R0 rates and PFS may at least partly reflect patient selection rather than a purely surgical approach effect.
Response01: We thank the reviewer for this important limitation and have performed additional analyses on risk of bias (Supp. Fig. F2/F4, updated Supp. Table S4). Further we now acknowlede and discuss the potential risk of selection bias already in the first paragraph of the discussion.
Comm02: Second, heterogeneity is substantial for several key outcomes (e.g., cytoreduction status, survival endpoints), yet its sources are only briefly mentioned. Where feasible, additional or more explicit subgroup/sensitivity analyses would strengthen the manuscript: for example, restricting the analysis to studies with propensity-score matching or low risk of bias, separating robotic from conventional laparoscopy, stratifying by study design (RCT vs. retrospective), or exploring the impact of surgical complexity and institutional volume. Even if such analyses are limited by the available data, a more detailed narrative discussion of heterogeneity and its implications for the certainty of pooled estimates would be valuable.
Response02: We agree that subgroup analyses strengthen the certainty of evidence and robustness of the findings. Thus we have included two subgroup analyses: 1) differentiation between studies with low to moderate risk of bias and high risk of bias; 2) studies including robotic surgery only in the MIS group versus mainly laparoscopy (no robotic) in the MIS cohort. Primary outcomes did not differ in both of these subgroup analyses, but level of significance decreased as expected with fewer patients per cohort. Results of these analyses are mentioned in the manuscript in the methods and results section, and depicted in Supplementary Figures F2.
Comm03: Third, the description and application of risk-of-bias and certainty-of-evidence tools could be clarified and better integrated into the interpretation of findings. It should be clearly stated which tool was used for randomized versus observational studies and how this influenced the GRADE rating for each outcome. Since many outcomes are ultimately rated as low or very low certainty, the language in the Abstract and Conclusions should be aligned with these levels; for example, phrasing the main message as “the available evidence suggests that MIS may be an oncologically safe alternative in selected patients, but the certainty of evidence is low and residual confounding cannot be excluded” would better reflect the current data.
Response03: We thank the reviewer for this comment. We have clarified the application of risk-of-bias and certainty-of-evidence tools in the methods section and have adapted the conclusion section as suggested to underline the risk for confounding.
Comm04: Fourth, the statistical methods would benefit from a more explicit description of how hazard ratios and standard deviations were derived when not directly reported. Indicating whether time-to-event data were estimated from Kaplan–Meier curves (e.g., using Tierney-type methods), specifying the formulas used to approximate SD from IQR or ranges, and briefly summarizing any sensitivity analyses (e.g., removing very large database studies, comparing fixed- versus random-effects models) would enhance reproducibility and allow readers to better judge the robustness of the results.
Response04: We approve with the reviewer and have specified the statistical approach in subsection 2.4 - Summary of the steps taken to derive the effect measures, as outlined in the Methods section:
Hazard ratios for overall and progression-free survival were determined by utilising the RevMan "Calculator" Action to compute raw data, such as SD and mean, reported in the original studies, into the logHR . No estimation from Kaplan–Meier curves or Tierney-type reconstruction methods was applied.
For continuous outcomes, missing standard deviations were derived solely from published 95% confidence intervals or interquartile ranges (IQRs) using Cochrane-recommended formulas (now explicitly reported). Binary outcomes were calculated from raw event data.
In addition we have included results of sensitivity analyses comparing fixed and random-effects models in the results section and Supplementary Figure F4.
Comm05: Finally, there are a few minor points that can improve clarity and readability. Some typographical errors and small inconsistencies in terminology should be corrected (e.g., “minimally invasie” → “minimally invasive”, “Wald-tpye” → “Wald-type”), and key definitions such as R0/R1 cytoreduction could be reiterated in table footnotes for quick reference. For the perioperative outcomes presented as standardized mean differences, it would help clinicians if the text and/or figure legends also highlight the approximate absolute differences in blood loss and length of stay. If available, a brief comment on histologic subtypes and their distribution (e.g., predominance of high-grade serous histology) would also help readers understand the generalizability of the findings.
Response05: Thank you for including these points in the report. We have corrected the typographical errors. To enhance clinical interpretability, we additionally report the corresponding absolute differences in the text (Chapter 3.4 Secondary Outcomes)

Reviewer 2 Report
Comments and Suggestions for Authors
Dear Authors,
Please find my comments on your manuscript directly on the PDF file.
Other comments:
Supplementary figures should be put in a supplementary document not uploaded separately and floating around.

Author Response
Comment01: Supplementary figures should be put in a supplementary document not uploaded separately and floating around.
Response01: We have compiled all supplementary figures (F1-F4) and tables (S1-S5) into a single word file to improve the overview.
all other comments and responses have been edited in the manuscript attached

Reviewer 3 Report
Comments and Suggestions for Authors
A review-type manuscript entitled ‘Outcome after Laparoscopic compared to Open Interval Debulking Surgery for Advanced Stage Ovarian Cancer: A Systematic Review and Meta-analysis’ was submitted by F. A.M. Saner and coworkers to be considered for its publication in the journal cancers by MDPI. Thus, after having read carefully the manuscript, I consider it suitable for the chosen journal because it fits into its scope, meeting well the journal’s standards. In addition, I found enough novelty and originality coming from the manuscript. The topic is interesting undoubtedly, and the timeframe covered is enough recent (10 last years). The paper is well written, easy to read, and the results are well discussed. The sample population study is statistically relevant (14 studies involving more than 16k patients). Section 4.7 is worth highlighting. The conclusion is shared with this reviewer because definitely I consider laparoscopy exhibits many advantages over laparotomy against advanced stage ovarian tumors. Thus, the manuscript is publishable practically in its present form. As unique suggestions: 1) the sharpness of figures 1, 2 and 3 must be substantially increased (compare, for instance, Figures 1, 2 and 3 (low sharpness) with table 1 (high-res)). 2) a non-sensitive cartoon-type comparison between laparoscopy and laparotomy in the introductory part (a new Fig. 1) will benefit the manuscript.
Author Response
Comment1
suggestions: 1) the sharpness of figures 1, 2 and 3 must be substantially increased (compare, for instance, Figures 1, 2 and 3 (low sharpness) with table 1 (high-res)). 2) a non-sensitive cartoon-type comparison between laparoscopy and laparotomy in the introductory part (a new Fig. 1) will benefit the manuscript.
Response1: We than the reviewer for this positive feedback. We have included higher resolution images of the figures and editable tables. As for meta-analyses, figure 1 is generally according to PRISMA guidelines the flowchart of study selection, we refrain from adding another figure 1 for clarity. However we agree to add a figure to visualize the aim, methods and outcome of the study (visual abstract) if desired to improve the perception of this meta-analysis.

Round 2
Reviewer 1 Report
Comments and Suggestions for Authors
The authors have adequately addressed all previous methodological and statistical concerns. The discussion has been strengthened with clearer acknowledgement of selection bias and heterogeneity, subgroup analyses were added, and the application of risk-of-bias and GRADE tools has been clarified. Minor wording and typographical issues have also been corrected. The revised version is scientifically sound, and I have no further essential comments. I support acceptance of the manuscript in its current form.
Author Response
We thank the reviewer for this positive evaluation and the relevant and helpful comments in round 1.
Reviewer 2 Report
Comments and Suggestions for Authors
dear authors,
thank you for the revisions. most the point by point responses were lacking which i suggest you submit next time with the revisions. not all revisors are so patient to go and check everything in parallel.
my last remarks: fix the arrows of the PRISMA figure. they're disconnected from the boxes
the CC0 CC1 - R0 discussion is worth having not to decrease the significance of your work but to highlight the need to analyze separately these 2 groups in future studies. If everyone continues with the minimal residue discussion, we will not progress beyond this, and careful stratification of these parients is needed.
best
Author Response
Comment 1: thank you for the revisions. most the point by point responses were lacking which i suggest you submit next time with the revisions. not all revisors are so patient to go and check everything in parallel.
Response 1: Please excuse the prior form of direct response to your comments in the tracked mode of the manuscript. As suggested we provide now a point by point response to the reviewer's comments of round 1 and round 2, please see the attachment.
Comment 2: fix the arrows of the PRISMA figure. they're disconnected from the boxes
Response 2: thank you for pointing out this graphical mistake, which we now corrected.
Comment 3: the CC0 CC1 - R0 discussion is worth having not to decrease the significance of your work but to highlight the need to analyze separately these 2 groups in future studies. If everyone continues with the minimal residue discussion, we will not progress beyond this, and careful stratification of these parients is needed.
Response 3: We thank the reviewer for raising this important conceptual issue. In our meta-analysis, most included studies reported cytoreduction either as “R0 vs. R0/R1” or combined R0 and R1 into a single “optimal” category, and only rarely provided sufficiently data to distinguish CC0 from CC1 across treatment groups. As a result, we were not able to stratify consistently by CC0 versus CC1/R1 and acknowledge this as a limitation of the current evidence base. In line with the reviewer’s suggestion, we added a paragraph in the Discussion/Strengths and limitations section explicitly addressing the ongoing shift toward CC0 as the preferred definition of optimal debulking and highlight that our pooled estimates mainly reflect the probability of achieving at most minimal residual disease (R0/R1) rather than strictly CC0/no macroscopic residual disease. We also point out that future trials comparing minimally invasive and open interval debulking should report and analyse CC0/R0 and CC1/R1 separately.
